# Evaluating Ultrasonicator Performance for Cyanobacteria Management at Freshwater Sources

**DOI:** 10.3390/toxins15030186

**Published:** 2023-03-01

**Authors:** Liam Vaughan, Dean Barnett, Elisa Bourke, Hamish Burrows, Fiona Robertson, Brad Smith, Jenna Cashmore, Michael Welk, Michael Burch, Arash Zamyadi

**Affiliations:** 1Water Research Australia, Adelaide, SA 5000, Australia; 2Intelligent Water Networks, Melbourne, VIC 3000, Australia; 3Central Highlands Water, Wendouree, VIC 3355, Australia; 4Goulburn Valley Water, Shepparton, VIC 3355, Australia; 5Australis Water Consulting, Adelaide, SA 5000, Australia; 6Department of Chemical Engineering, The University of Melbourne, Parkville, VIC 3010, Australia; 7Department of Civil Engineering, Monash University, Clayton, VIC 3800, Australia

**Keywords:** cyanobacteria, drinking water, reservoir management, sonication, ultrasonication, non-chemical dosing

## Abstract

Algal blooms consisting of potentially toxic cyanobacteria are a growing source water management challenge faced by water utilities globally. Commercially available sonication devices are designed to mitigate this challenge by targeting cyanobacteria-specific cellular features and aim to inhibit cyanobacterial growth within water bodies. There is limited available literature evaluating this technology; therefore, a sonication trial was conducted in a drinking water reservoir within regional Victoria, Australia across an 18-month period using one device. The trial reservoir, referred to as Reservoir C, is the final reservoir in a local network of reservoirs managed by a regional water utility. Sonicator efficacy was evaluated through qualitative and quantitative analysis of algal and cyanobacterial trends within Reservoir C and surrounding reservoirs using field data collected across three years preceding the trial and during the 18-month duration of the trial. Qualitative assessment revealed a slight increase in eukaryotic algal growth within Reservoir C following device installation, which is likely due to local environmental factors such as rainfall-driven nutrient influx. Post-sonication quantities of cyanobacteria remained relatively consistent, which may indicate that the device was able to counteract favorable phytoplankton growth conditions. Qualitative assessments also revealed minimal prevalence variations of the dominant cyanobacterial species within the reservoir following trial initiation. Since the dominant species were potential toxin producers, there is no strong evidence that sonication altered Reservoir C’s water risk profiles during this trial. Statistical analysis of samples collected within the reservoir and from the intake pipe to the associated treatment plant supported qualitative observations and revealed a significant elevation in eukaryotic algal cell counts during bloom and non-bloom periods post-installation. Corresponding cyanobacteria biovolumes and cell counts revealed that no significant changes occurred, excluding a significant decrease in bloom season cell counts measured within the treatment plant intake pipe and a significant increase in non-bloom season biovolumes and cell counts as measured within the reservoir. One technical disruption occurred during the trial; however, this had no notable impacts on cyanobacterial prevalence. Acknowledging the limitations of the experimental conditions, data and observations from this trial indicate there is no strong evidence that sonication significantly reduced cyanobacteria occurrence within Reservoir C.

## 1. Introduction

### 1.1. Harmful Algal Blooms and Source Water Management

Management of freshwater bodies including lakes, rivers, and reservoirs is becoming increasingly challenging with the proliferation of harmful algal blooms (HABs), which are often comprised of potentially toxic cyanobacteria [1]. A survey conducted across the United States, Australia, and Canada found that this challenge is not geographically isolated, with all 35 surveyed water utilities reporting pelagic cyanobacteria blooms [2]. Both local and global factors contribute to cyanobacterial growth, particularly increased temperatures, increased mean water residence time, and increased influx of nutrients from surrounding agricultural and urban activities [3,4,5]. Public health risks primarily arise when toxic cyanobacterial growth occurs within source water, such as drinking water reservoirs, where cyanotoxins may cause adverse health impacts [6]. Beyond safeguarding public health, waterbody management aims to mitigate detrimental social, environmental, and economic outcomes [7,8]. Non-toxic cyanobacterial growth can still be disruptive for waterbody management by introducing unpleasant taste and odor (T&O) compounds, lowering oxygen levels, impacting water appearance, and raising treatment costs [9]. A low influx of cyanobacteria cells into a drinking water treatment plant (DWTP) can also cause accumulation and thus risk the safety and quality of treated water regardless of HAB presence within source water [10].

Several techniques and technologies have been implemented within source waterbodies for HAB management, from prevention, in-waterbody treatment of ongoing blooms, and DWTP treatment [5,11,12]. Watershed management is at the center of a preventative management approach and seeks to lower cyanobacterial biomass drivers by limiting nutrient runoff from surrounding areas, such as in the case of agricultural fertilizer use [2,13]. Capping agents, such as lanthanum-activated modified clay and alum, can prevent sedimentary phosphorus from leaching into water; however, these agents may produce water quality issues and the associated ecological risks are poorly understood [14]. Dosing waterbodies with nutrient sequestering chemicals presents an additional option for active source water control [15]. Kibuye et al. found that among surveyed utilities, algaecide addition and aeration were the two most frequently deployed source water management strategies [2]. Chemical dosing can be an effective short-term strategy for ongoing HAB impact minimization; however, long-term algaecide use, such as with copper-based compounds, may cause eco-toxicity in other non-target organisms [16,17]. Non-copper-based algaecides, such as hydrogen peroxide, are also receiving growing attention as an alternative to traditional chemical dosing [5]. Peroxide dosing has been found to effectively induce oxidative stress in cyanobacterial cells; however, a longer lag time has been observed prior to full efficacy relative to copper-based algaecides [14]. Mechanical agitation using equipment like a surface mixer or aerator can cause destratification of water columns and has exhibited localized light-limiting impacts on algae at shallow depths and within 5 meters of the device [14,18]. Due to its limited range, this technique’s efficacy is considered low on a reservoir scale [14]. Kibuye et al. outline a comprehensive review of the discussed techniques and additional approaches [2,18]. 

### 1.2. Sonication

Sonication (or ultrasonication) is a non-chemical preventative cyanobacteria management technique which aims to remove cells from water columns without inadvertently harming local ecology or introducing cyanobacterial metabolites into water. Cyanobacteria use gas vesicles to regulate their buoyancy, allowing them to migrate vertically through waterbodies and thereby access desired nutrient, light, and temperature levels [5,16]. Sonication devices target cyanobacteria by inducing cavitation within these vesicles, whereby localized regions of high temperature and pressure are generated, which can cause collapse [19]. Sonication efficacy is influenced by vesicle characteristics such as wall strength and size, so the sonic frequency and power density required to achieve vesicle collapse differ between species, requiring device settings to be adapted accordingly [16,20]. The direct targeting of these vesicles means that sonication selectively inhibits cyanobacteria over other microorganisms which lack these cellular nano-compartments [19]. Sonication-induced reductions in photosynthetic capabilities are another potential mechanism for cyanobacterial inhibition and occur due to phycocyanin structural damage [16]. Induced cavitation can also cause colony de-clumping or disruption of filamentous structures, which reduces the survivability of affected cells [19,21]. The effective range of ultrasound propagation is governed by the inverse square law, whereby induced pressure levels halve with a doubling of distance. Turbid water conditions can also inhibit wave propagation, potentially reducing sonication efficacy within bloom-laden source water [21]. Level of exposure to sonication is usually determined by the electrical energy supplied to transducers, exposure duration, acoustic energy inputted into water, average sound intensity, and power density [16].

While limited trials have been conducted into non-desirable impacts on non-target organisms, risks appear to be low. Keijzers and Postma examined the effects of sonication on zooplankton during a sonicator trial in the Netherlands’ Zoetermeerseplas. Reproduction and survival of *Daphnia magna* were studied within this field trial, and no detrimental impacts were observed regardless of proximity to the device [22].

While treatment units are commercially available, published field trials of sonication is limited and most sonication studies have been conducted in laboratory contexts with a limited variety of taxa and conditions. Field trials for drinking water source management are even more limited [16,20,21]. These trials have had mixed outcomes and there have been limited applications within drinking water reservoirs, which has led many utilities to question the efficacy of this technology [21]. 

Hobson et al. tested sonicator efficacy within three trials: at a laboratory scale (20 L) and within a larger pond (7000 L) over a 35-day period and an ultrasound/ozonation device within a lake setting. Sonication proved ineffective within the laboratory and larger pond trials, and the sonication/ozonation device was also unable to reduce cell numbers within a *Cylindrospermopsis* bloom or affect cell viability [14]. Purcell et al. conducted a five-month sonication trial on a waterbody with an area of 0.14–0.18 km^2^ and an average depth of 4.3 m using 40 W, 40–50 kHz transducers. This trial found no significant variations between test and control sites [23]. 

Other trials have yielded more positive results. Sonicator devices were trialed within two separate wastewater balance tanks at an Australian wastewater treatment plant. These waterbodies were considerably larger than a laboratory scale trial but smaller than a reservoir. Sonication was found to effectively inhibit biofilm growth due to turbulence induced by the sonication, and during the trial there was a reduced requirement for backwash, which indicated a lower influx of cells from the balance tanks. A lag time of four weeks from device installation to effective cell density reduction was observed. Due to the trial’s location within an operational water treatment system, necessary disruptions such as vessel cleaning and high nutrient carryover events occurred, which impacted the ability for operators to determine device efficacy. Despite this, the consensus was that the devices were effective [24]. Schneider et al. trialed four sonicator buoys within the Canoe Brook Reservoir 1 in New Jersey, USA, over a 5-month period from spring to summer. This reservoir, with a volume of 3.44 million m^3^, is often eutrophic due to high nutrient and mineral loads. Operators observed effective control of algae and T&O compounds when correct control schemes were used. Unlike the device used within this trial, output frequencies required manual adjustment, which was used to control the emergence of an *Aphanizomenon* bloom that subsequently declined after settings were changed. Operators noted a 22% reduction in alum dosing requirements relative to the same period in the previous year, lower turbidity of dissolved air floatation effluent, and an 83% increase in unit filter run volumes. It was determined that this system provided USD 87,800 (2014 dollars) in savings with a payback time of 1.8 years [20]. Park et al. trialed sonication at 36 kHz and 300 W, 108 kHz and 450 W, and 175 kHz and 650 W. This resulted in a short-term reduction in cell density; however, researchers also detected localized release of toxins from lysed cells [25]. 

Reviewed trials illustrate the challenges faced by efforts to validate sonication. Many are conducted within functional water infrastructure which must prioritize functionality for consumers over experimental validity. Others were conducted within non-realistic hydrological systems that do not reflect reservoir water settings accurately. Other trials relied primarily on qualitative observations, while few were able to correct for the multitude of potential influencing factors, such as climatic conditions and anthropogenic inputs. An additional commonly observed theme within published trials pertained to device reliability. During the Australian wastewater treatment plant trial, the transducer on a sonication unit failed and prompted operators to hold an additional unit as a spare in case of subsequent failures [24]. Schneider et al. also experienced technical difficulties when their master buoy failed and cell counts subsequently rose [20]. Hence, the aim of this study was to evaluate the performance of ultrasonication technology for cyanobacteria management at freshwater sources.

## 2. Materials and Methods

### 2.1. Technology Selection and Implementation

As an emerging market, ultrasonicator supplier selection was performed based on initial reliability and availability. The selected supplier’s products were immediately available for implementation within Victoria, Australia, facilitating prompt trial initiation. As discussed previously, in addition to technology evaluation, the supplier was subject to evaluation within this trial to provide insights into challenges that may be faced during potential future engagements. The supplier has requested to remain anonymous within this paper.

In accordance with supplier advice, a single solar-powered sonicator unit was installed in Reservoir C on 3 December 2020. The supplier advised that this device with two transponders (using 100 watts of power, the transducers send out ultrasound across a wide frequency range—from 20 to 60 KHz) was sufficient to cover the volume of Reservoir C (a small reservoir of 350–400 megaliters). Selecting an appropriate number of transponders based on the target waterbody is a key design factor. 

### 2.2. Site Selection and Sampling Protocol

Reservoir C is the final drinking water reservoir in a system of connected reservoirs (Figure 1 and Figure 2). Reservoir C is operated to provide water to an associated DWTP which subsequently supplies the local district and surrounding communities. Water is transported from Reservoirs A, B, and D to Reservoir C as required to optimize the water supply.

Sampling was conducted according to the local utility’s routine historic and ongoing sampling program in triplicate [6]. Two sampling points were available at Reservoir C (‘DWTP intake’ and ‘reservoir water’), while one point (‘reservoir water’) was available at Reservoirs A, B, and D. At Reservoir C, ‘reservoir water’ samples were taken as weekly to fortnightly grab samples at an approximately 30 cm depth at the end of the reservoir walkway, a structure extending from the reservoir shore above the DWTP intake pipe (Figure 1). ‘DWTP intake’ sampling was conducted from this pipe to provide an accurate representation of water quality abstracted from the reservoir and entering the treatment plant (Figure 1). Grab samples for phytoplankton analysis were collected within 1-liter opaque plastic bottles.

Samples from both locations were transported at between 2 and 8 °C to the laboratory for algal and cyanobacterial identification and quantification, with delivery targeted within 8 h [6]. ‘Reservoir water’ samples from Reservoirs A, B, and D were taken according to the same protocols as observed for Reservoir C. Cyanobacteria biovolumes (BVs) and cell counts were recorded to the species level, while green algae, flagellates, and diatoms had only their cell counts recorded [6]. Additional samples were taken from geographically proximate reservoirs operated by the local utility (Figure 2); however, the sampling frequency at these locations was lower than Reservoir C. 

Regular temperature data were collected by a YSI V6 fluorometer (Yellow Spring Instrument, Yellow Spring, OH, USA) probe situated at the Reservoir C walkway. Sample metadata and chain of custody information, including the type of sample collected, sampling date, location, and time, were included. In case of device malfunction, the period affected was also noted. Results from routine sampling were made available courtesy of the local utility for the three years prior to sonicator installation and until trial completion at the end of May 2022.

Logistical challenges complicated efforts to implement a true control. An ideal control would have required a water curtain across the tested reservoir, with sampling occurring on both sides and sonication occurring on one. Due to the reservoir’s operational requirements of supplying drinking water to local communities, this was not practical within the scope of the study. A quasi-control was instead implemented whereby data from pre-sonication periods were evaluated relative to post-sonication data. Data could also be cross-checked with climate data and the samples collected from adjacent non-trial reservoirs to identify potential broader climate impacts. The local climate around the trial site was approximated using Bureau of Meteorology historical data of monthly rainfall and mean maximum monthly temperatures. As discussed within this paper, these imperfect controls have implications on the reliability of conducted statistical tests. 

Additional qualitative operator observations were also recorded and considered when determining sonicator efficacy, such as the presence of surface scums and technical interruptions.

### 2.3. Data Consolidation and Manipulation

Data from all sites were collated into Microsoft Excel according to sampling time and location for subsequent analysis. Data from each site were separated approximately into bloom and non-bloom conditions based on YSI temperature readings. Historical observations at the reservoir and reviewed literature indicate blooms are unlikely at water temperatures below 15 °C, which occurs between the beginning of May and the end of September. While it must be acknowledged that some species can thrive at temperatures below 15 °C, lower temperatures severely inhibit dominant reservoir species such as *Microcystis* [26,27]. Datasets were therefore divided into bloom (October–April) and non-bloom (May to September) seasons (Table 1). Note that due to the lower sampling frequency maintained by operators during cooler months, the non-bloom datasets are considerably smaller than bloom counterparts.

Timeseries data were qualitatively examined against the World Health Organization’s Alert Level 1 (cell count ≥ 2000 cells/mL, biovolume ≥ 0.2 mm^3^/L) and Alert Level 2 (cell count ≥ 100,000 cells/mL, biovolume ≥ 10 mm^3^/L) as a means of benchmarking source water risk and the impact that sonication potentially had on this risk, if any. 

For species-level breakdowns, repetitions were consolidated into a single species to provide a more accurate representation of overall community compositions (e.g., ‘*Dolichospermum Coiled*–Small < 6 μm’ and ‘*Dolichospermum Coiled*–Large > 6 μm’ were consolidated to ‘*Dolichospermum*’). Dominant species according to biovolumes and cell counts were then plotted on a timeseries to determine the impact sonication had on these species.

### 2.4. Technology Evaluation

A statistical analysis was conducted between pre- and post-sonication periods within each dataset, separated into the previously discussed bloom and non-bloom periods. Biological datasets often exhibit lognormality, so lognormality within the studied data was examined [28]. A natural logarithm was applied to each cell count and biovolume value from which a histogram was constructed for each dataset to determine if the discrete datapoints approximated a lognormal distribution. The log-means of pre- and post-sonication populations were computed and a heteroscedastic *t*-test was conducted. Homoscedasticity could not be assumed because the impact of sonication on post-sonication population variance is unknown. A two-tailed test was used to account for positive or negative differences in population means.

Hypothesis testing was used to test for statistical differences in pre- and post-sonication populations with a significance of α = 5%. A null hypothesis was tested whereby the difference between population means was hypothesized to be zero (i.e., X: H_0_ = 0), with an alternative hypothesis that the difference was not equal to zero (i.e., X: H_1_ ≠ 0). The null hypothesis was rejected when *t*-tests yielded *p* < 0.05, indicating that there was a statistical difference between pre- and post-sonication population means in either a positive or negative direction.

Additional qualitative evaluations were made based upon operator observations for further evaluation of the used technology.

## 3. Results and Discussion

### 3.1. Climate Comparison

Analysis of climate data across the trial period reveals substantial variability (Figure 3). The region received considerably less rainfall in 2018 (total = 417.9 mm) and 2019 (total = 348.6 mm) than 2020 (total = 510.5 mm) and 2021 (total = 610.7 mm). The average rainfall per month during the post-sonication period was 29.9% higher than the pre-sonication period. Rainfall generally facilitates favorable cyanobacteria growth conditions as runoff transports nutrients from surrounding areas into reservoirs [29]. High-rainfall events can also have opposite impacts on cyanobacteria cell densities depending on the hydrodynamics of the studied reservoir system [30]. Since Reservoir C is supplied by proximate reservoirs rather than its own catchment, a rainfall event is unlikely to elevate cyanobacteria levels until water from feeder reservoirs, which have greater exposure to agricultural activity and therefore nutrient inputs, reaches Reservoir C. This may have a lengthy and non-constant lag time, which complicates efforts to decouple rainfall variability from algal prevalence. The aggregate average temperature from pre-sonication months was 0.5 °C higher than post-sonication months; however, water temperature data are not available for the entire period. Cyanobacterial growth is generally optimal at warmer water temperatures of 25 °C or greater [26,30].

### 3.2. Algal Groups Timeseries—Reservoir C

A logarithmic timeseries depicting cell counts of all algal groups as measured within the DWTP intake pipe was plotted (Figure 4) to facilitate qualitative observations across the entire dataset. Cyanobacteria cell counts measured prior to sonicator introduction were relatively low, which makes it challenging to observe a clear inhibition effect. Cyanobacteria cell counts exceeded Alert Level 1 each bloom season before and after sonicator installation, with all maximum ‘post-sonication’ bloom season cell count values falling approximately within 0.5 orders of magnitude relative to maximum ‘pre-sonication’ bloom season cell counts, except for a single extreme value in early 2018. Examining non-cyanobacteria species reveals a notable increase in the 2021/2022 bloom season relative to earlier seasons, which likely indicates favorable microbial growth conditions during this season. A lack of a corresponding increase in cyanobacteria cell counts may be evidence of inhibitory effects from sonication counteracting these conditions, or it may simply indicate that this change in conditions did not benefit cyanobacteria. Examining the Reservoir C cell count timeseries using reservoir water data (Appendix A) reveals a similar rise in non-cyanobacteria species with no corresponding cyanobacteria rise. A similar examination of geographically proximate reservoirs (Appendix A), however, reveals a lack of a corresponding trend for cyanobacteria or other algal species, suggesting the observed trends in Reservoir C are due to local factors which cannot be definitively identified using available data.

### 3.3. Cyanobacteria Timeseries—Reservoir C

Figure 5 depicts the cyanobacteria biovolumes corresponding to cell counts depicted in Figure 4 and Appendix A. Data from the DWTP intake and reservoir water sets were plotted on the same graph to determine differences between each sampling location, which may allude to spatial differences in sonicator efficacy. The key difference is an apparent increase in extreme values, as measured by reservoir water three-sample moving averages relative to DWTP intake moving averages; however, this is likely due to the higher sampling frequency at the DWTP intake, which serves to smooth more extreme values. A qualitative analysis of both biovolume timeseries before and after sonicator installation positions the 2021/2022 bloom season as unremarkable—lower biovolumes were generally recorded relative to 2017/2018, higher biovolumes were generally recorded relative to the 2018/2019 season, while similar values were recorded in 2019/2020. It must be noted that as with the previously discussed cell counts, cyanobacteria biovolumes at Reservoir C rarely exceeded Alert Level 1, which makes any sonicator-induced biovolume reductions harder to detect.

### 3.4. Cyanobacteria Timeseries—All Reservoirs

Cyanobacteria biovolumes at each monitored reservoir were considered (Figure 6) to check for broader geographical trends that may be indicative of climatic influences. It can be immediately noted that Reservoir C generally had lower cyanobacteria biovolumes than other reservoirs prior to sonicator installation. The gap between reservoirs appears to narrow following sonicator installation, which is likely due to favorable growth conditions within 2021/2022 unique to Reservoir C. This observation also implies that the sonicator was incapable of fully counteracting these conditions.

### 3.5. Reservoir C Cyanobacteria Composition

The dominant four species on a cell count (Figure 7) and biovolume (Figure 8) basis were plotted to check for disturbances in the composition of these species following sonicator installation. The top three dominant species according to each measurement type (cell count or biovolume) were subsequently plotted to illustrate total reservoir diversity (Figure 9). On a cell count basis, species which were historically dominant remained dominant following sonicator installation, except for *Synechocystis* sp., which has been rarely detected within reservoir water (Appendix A) or DWTP intake sampling since installation. On a biovolume basis, recent and historical cyanobacterial growth has largely consisted of *Microcystis*, *Dolichospermum*, and *Aphanizomenon*. Since these species are also potential cyanotoxin producers, a lack of disruption to their prevalence following sonicator installation indicates that the sonicator did not effectively reduce reservoir water risk profiles.

### 3.6. Sonicator Efficacy—Statistical Evaluation

Datasets analyzed within the trial were found to fit lognormal distributions. An example is provided in Figure 10. Datasets with a smaller number of observations (e.g., non-bloom periods and non-trial reservoirs) had a weaker fit; however, this was attributed to the smaller sample sizes and was therefore taken to also demonstrate lognormality. All datasets were then analyzed for statistical differences between pre- and post-sonication populations (Table 2).

Analysis of DWTP intake sampling showed no statistically significant change when cyanobacteria biovolumes were compared between pre- and post-sonication periods. This sampling did, however, show a statistically significant reduction in cyanobacteria cell counts during bloom seasons. As previously identified qualitatively, DWTP intake sampling also revealed a general increase in other algal groups’ cell counts (a significant increase in green algal and flagellate CCs across bloom and non-bloom periods and a significant increase in diatoms in non-bloom periods) following sonicator installation, which may be indicative of locally favorable conditions for algal growth. The sonicator may have prevented proportional growth of cyanobacteria in Reservoir C; however, this cannot be further verified without comprehensive water quality data, which were not available to the authors of this paper. Since eukaryotic algal species are less problematic in terms of complicating water treatment, a trial outcome in which cyanobacterial prevalence is replaced with other algal species is considered successful from a water management perspective. A definitive link cannot, however, be established based on this trial’s observations. As previously identified, climatic variations were likely to have had some impact on the validity of this analysis.

Analysis of reservoir water sampling within Reservoir C revealed no statistically significant change in cyanobacteria presence during bloom seasons following sonicator installation. A significant increase in cyanobacteria BVs and CCs was detected during non-bloom seasons. Reservoir water sampling supported DWTP intake observations of increasing non-cyanobacteria algal cell counts after sonicator installation. This analysis again faces reliability issues, as local weather was not consistent across the period, and it also involves datasets that are smaller than the DWTP intake datasets.

Other statistically significant changes were observed across the period within populations that were not affected by the trial (i.e., those in Reservoirs A, B, and D). While the number of significant changes within the other reservoirs was lower than those observed in Reservoir C, this highlights the potential for the changes observed to have been caused by broader environmental variations. Comparing the trial and non-trial locations directly, the general increase in algal activity that was observed in Reservoir C after sonicator installation was not observed within non-trial reservoirs during the same period. The validity of the analyses conducted for other reservoirs is even lower than those conducted using Reservoir C data due to the small sample sizes involved.

A considerable limitation to trial validity is the previously discussed input of water to Reservoir C from surrounding reservoirs. Volumetric flowrate into Reservoir C from A, B, and D is non-constant and is selected based on algal and water quality observations within the feeder reservoirs. The resulting influx of algae and cyanobacteria from feeder reservoirs is therefore low but non-zero and varies over time.

A further limitation to this analysis is the poorly understood lag time between sonicator installation/activation and effective cyanobacterial inhibition. Due to the limited trial period of 18 months, it was, however, considered important for all data collected following sonicator installation to be included in this analysis. This is further justified when it is considered that the time before effective treatment is a key consideration when considering the efficacy of source water control methods, so if a lengthy lag time precedes effective cyanobacteria inhibition, it should be factored into an efficacy assessment depending on device use within a reactive or preventative role. A potential method for lag time impact mitigation within future analyses may be device installation during non-bloom periods (e.g., the middle of winter). This would allow the corresponding non-bloom period to be eliminated from data analysis without limiting total data availability.

### 3.7. Sonicator Efficacy—Operator Observations

As with other trials reviewed within this report’s literature review, the sonicator trial and Reservoir C also experienced technical difficulties. Shortly after device installation, one of the device’s two transponders failed for a period of approximately two weeks. The exact duration of this outage is unknown, as operators were not alerted that technical issues had been encountered. It must be noted that the supplier was responsible for monitoring and raising alarms. Analysis of data during this period was unable to detect any tangible changes in cyanobacteria cell counts or biovolumes following this failure. 

Later during the trial, the device failure alarm was sounded. The supplier investigated this and subsequently reported that it had been a false alarm and that the device was operating correctly. 

## 4. Conclusions

An 18-month sonication trial was conducted at Reservoir C involving one sonicator device. Qualitative analysis of general algal and cyanobacterial trends within Reservoir C and surrounding reservoirs across the years preceding and following the trial revealed a slight increase in *eukaryotic* algal growth within Reservoir C following device installation and no strong evidence of a change in cyanobacteria densities. A qualitative species-level analysis revealed minimal disruptions to the dominant cyanobacteria species within the reservoir following trial initiation. A subsequent statistical analysis primarily revealed a significant elevation in *eukaryotic* algal cell counts in the bloom and non-bloom periods following installation. Statistical analysis of cyanobacteria datasets collected at Reservoir C primarily revealed no significant changes, excluding a significant bloom season cell count reduction measured within the DWTP intake pipe and a significant non-bloom season biovolume and cell count increase measured within the reservoir. Assessment of operator observations revealed some communication challenges with the supplier and technical difficulties during the trial. While there was no conclusive evidence that sonication was able to significantly reduce cyanobacteria cell counts across the trial duration (Appendix A), observations indicate that optimizations in technology and supplier reliability are necessary.

## Figures and Tables

**Figure 1 toxins-15-00186-f001:**
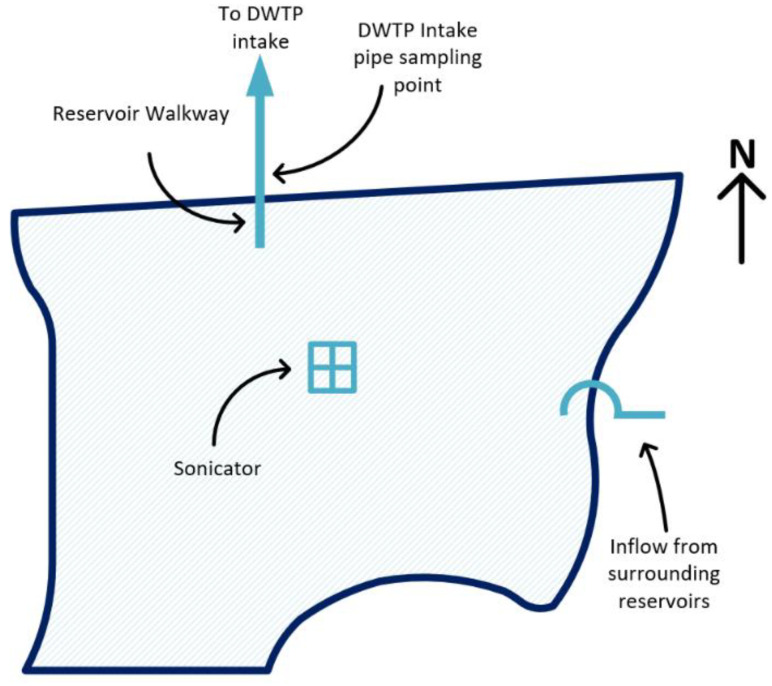
Reservoir C map. Figure not to scale.

**Figure 2 toxins-15-00186-f002:**
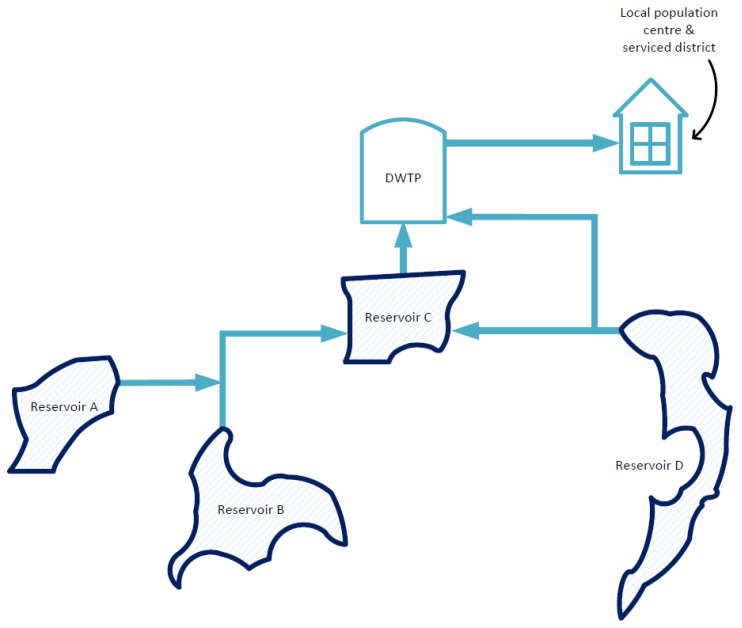
Simplified schematic of the local utility’s serviced district supply system. Figure not to scale.

**Figure 3 toxins-15-00186-f003:**
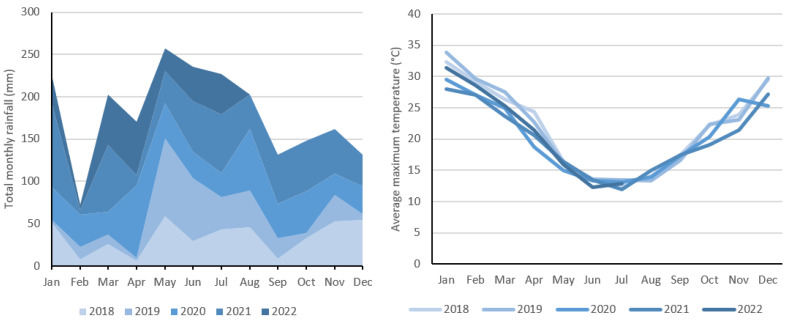
Climate data comparison from Jan 2018 to Jul 2022. (**Left**) Total monthly rainfall for each trial year. (**Right**) Mean monthly maximum temperature for each trial year. Data retrieved from Bureau of Meteorology (2022).

**Figure 4 toxins-15-00186-f004:**
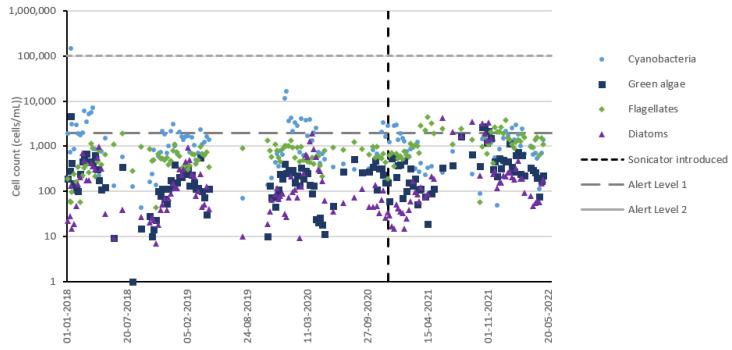
Reservoir C all algal groups timeseries using drinking water treatment plant (DWTP) intake dataset.

**Figure 5 toxins-15-00186-f005:**
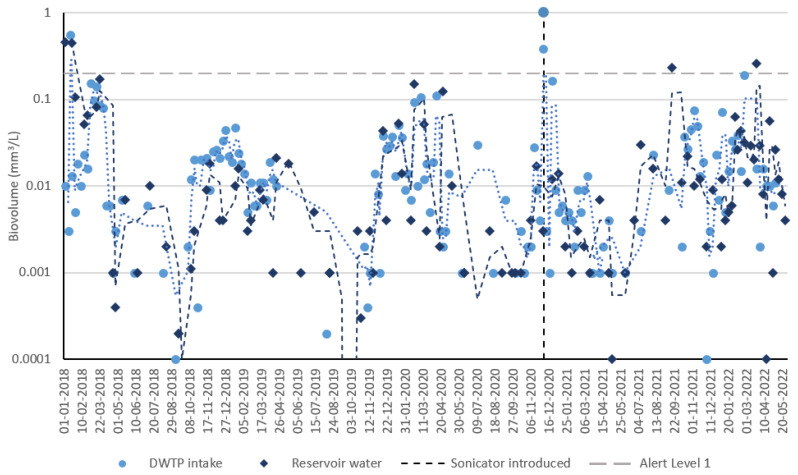
Reservoir C reservoir cyanobacteria biovolume timeseries using reservoir water and DWTP intake datasets. Dotted lines represent a moving average of the three most recent observations from each dataset (light blue is DWTP intake data; dark blue is reservoir water data).

**Figure 6 toxins-15-00186-f006:**
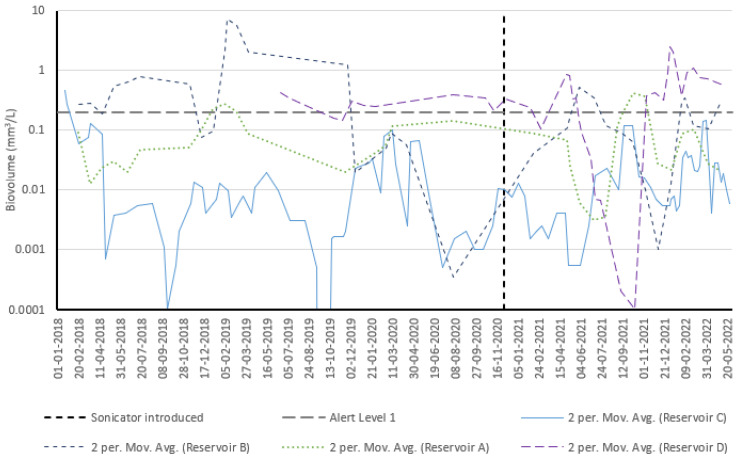
Cyanobacteria biovolume timeseries across all reservoirs (reservoir water dataset used from all locations).

**Figure 7 toxins-15-00186-f007:**
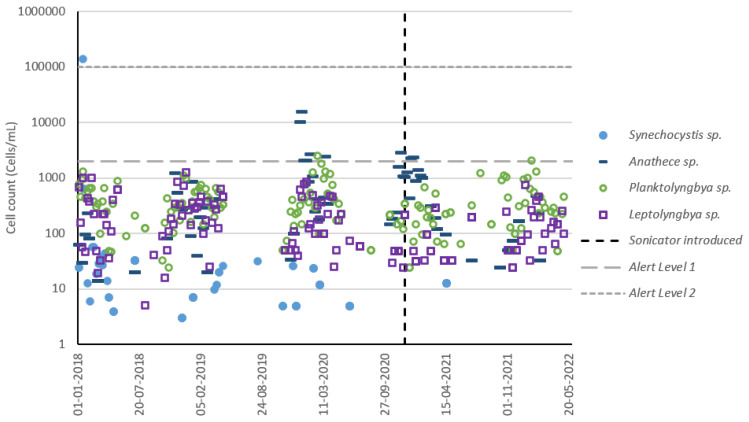
Reservoir C cyanobacteria cell count (dominant species) timeseries using DWTP intake dataset. Top four dominant species determined based upon cell count averages across entire dataset.

**Figure 8 toxins-15-00186-f008:**
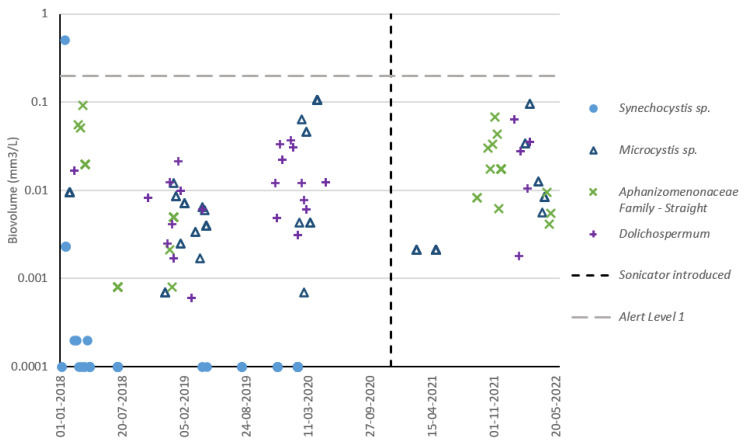
Reservoir C cyanobacteria biovolume (individual species) timeseries using DWTP intake dataset. Top four dominant species determined based upon biovolume averages across entire dataset.

**Figure 9 toxins-15-00186-f009:**
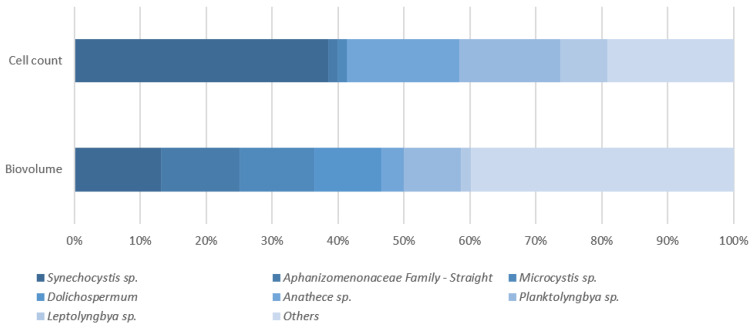
Reservoir C cyanobacteria species breakdown using DWTP intake data.

**Figure 10 toxins-15-00186-f010:**
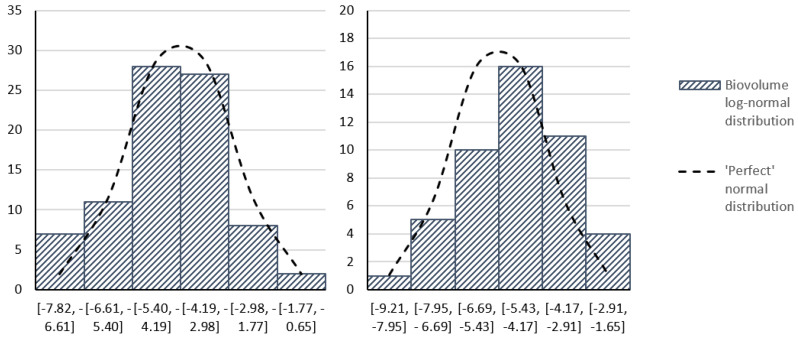
Example qualitative evaluation of lognormality. Both figures depict Reservoir C cyanobacteria biovolumes from the bloom season dataset (sampled at DWTP intake) transformed into a lognormal distribution and compared with an ideal normal distribution. (**Left**) Pre-sonicator installation; (**Right**) post-sonicator installation.

**Table 1 toxins-15-00186-t001:** Sampled locations and dataset sizes, where green-shaded cells indicate a larger dataset and red-shaded cells indicate a smaller dataset relative to others.

Location	Data Source	Season	n (Pre)	n (Post)
Reservoir C	DWTP intake	Bloom	83	47
		Non-bloom	11	6
	Reservoir water	Bloom	42	33
		Non-bloom	18	11
Reservoir A	Reservoir water	Bloom	14	8
		Non-bloom	4	8
Reservoir B	Reservoir water	Bloom	16	9
		Non-bloom	4	8
Reservoir D	Reservoir water	Bloom	8	16
		Non-bloom	3	10

**Table 2 toxins-15-00186-t002:** Statistical analysis results. Deeper-green-shaded cells indicate a larger log reduction in average algal densities pre- and post-sonication, while deeper-red-shaded cells indicate larger increases. Cells with an ‘X’ indicate a statistically significant change according to hypothesis testing.

Location	Data Source	Season	Cyanobacteria (BV)	Cyanobacteria (CC)	Green Algae	Flagellates	Diatoms
Reservoir C	DWTP intake	Bloom		X	X	X	
		Non-bloom			X	X	X
	Reservoir water	Bloom			X	X	
		Non-bloom	X	X	X		X
Reservoir A	Reservoir water	Bloom					
		Non-bloom				X	
Reservoir B	Reservoir water	Bloom					
		Non-bloom					
Reservoir D	Reservoir water	Bloom					
		Non-bloom	X	X			X

## Data Availability

Data are available on request.

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
