# Peer review of "Evaluating Ultrasonicator Performance for Cyanobacteria Management at Freshwater Sources"

_toxins, 2023, doi:10.3390/toxins15030186_

Round 1

Reviewer 1 Report

The authors pay attention to the adverse effects and challenges of harmful algal blooms on global freshwater management, and design field experiments to evaluate the ultrasonicator performance for cyanobacteria management at the source. Through qualitative and quantitative analysis of algal and cyanobacterial trends, the authors evaluate sonicator efficacy and finally indicate that sonication cant significantly reduce the occurrence of cyanobacteria. This manuscript is interesting and meaningful, but it is not acceptable at this time. There are still some crucial points to be solved, I would like to point out some of my notes as follows:

1. In my opinion, the authors should introduce some contents about the cyanobacterial harmful algal blooms and specific toxin producing taxa in the history of Reservoir A, B, C and D.

2. 1.1. Harmful algal blooms and source water management-...of harmful algal blooms (HABs)which are... should be ...of harmful algal blooms (HABs) which are... , please correct it.

3. Where is the specific location of the site “reservoir water? Please provide. 

4. What is the specific detection method of DWTP intake, including physical and chemical parameters, phytoplankton and cyanobacterial counts, etc.

5. For cyanobacteria biovolumes (BV) and cell counts method, the description is too simple. What is the sampling frequency? What is the volume of water sample used for qualitative observation and quantitative counting? What are the specific operation steps? Is each quantitative sample counted three times to eliminate the error?

6. Environmental factors in Reservoir C are not presented in this paper, and the nutritional levels of these reservoirs are also unknown.

7. There are two 3.2 contents in the result section (3.2. Algal groups timeseries – Reservoir C and 3.2. Cyanobacteria timeseries – Reservoir C), please check and correct.

8. Phytoplankton and cyanobacterial community composition, diversity and richness are also unknown, because I cant find the specific data. I think that dominant species and rare species should be included in the results.

9. Whether there is a significant difference should be marked at the corresponding position in the text, such as (p<0.05/0.01).

10. The presentation of all figures can be improved. And figures 7-9: Latin names of species should be italicized. Some figures can focus on the main contents, such as S1, S5 and S6. I suggest that the data after 2018 should be presented separately. These figures cannot be seen clearly.

11. What's percentage of toxic cyanobacteria in different sites and at different sampling periods? How about the species diversity before and after ultrasonic treatment, was it a single strain or mixed strains? I suggest using a figure or table to make a clear summary.  

Finally, I think this paper didnt show the specific results of toxins, or whether cyanotoxins were detected in the reservoirs? How about the dynamics of species abundance and toxin-producing before and after Sonicator treatment? I am very interested in this point. I hope the authors can elaborate on the community composition and dynamic changes of harmful cyanobacteria, as well as the concentration and dynamic changes of cyanotoxins.

Author Response

Response to Reviewer 1:

The authors pay attention to the adverse effects and challenges of harmful algal blooms on global freshwater management, and design field experiments to evaluate the ultrasonicator performance for cyanobacteria management at the source. Through qualitative and quantitative analysis of algal and cyanobacterial trends, the authors evaluate sonicator efficacy and finally indicate that sonication can’t significantly reduce the occurrence of cyanobacteria. This manuscript is interesting and meaningful, but it is not acceptable at this time. There are still some crucial points to be solved, I would like to point out some of my notes as follows:

The authors acknowledge the reviewer’s comments and have addressed them all as listed below to improve the manuscript for publication.

  1. In my opinion, the authors should introduce some contents about the cyanobacterial harmful algal blooms and specific toxin producing taxa in the history of Reservoir A, B, C and D.

The authors acknowledge the reviewer’s comments and would like to highlight the presentation of historic data from 2018 as in Figure 3. Also, please note majority of historic data are based on visual observation rather than taxonomic speciation. 

  1. “1.1. Harmful algal blooms and source water management-...of harmful algal blooms (HABs)which are...” should be “...of harmful algal blooms (HABs) which are...” , please correct it.

The authors acknowledge the reviewer’s comments, and the text has been modified as suggested.

  1. Where is the specific location of the site “reservoir water”? Please provide.

Is the water inside the reservoir and each reservoir has been identified separately (Figures 1 and 2).

  1. What is the specific detection method of DWTP intake, including physical and chemical parameters, phytoplankton and cyanobacterial counts, etc.

The authors acknowledge the reviewer’s comments and would like to highlight that same detection methods was used for all sampling sites as indicated within the manuscript.

  1. For cyanobacteria biovolumes (BV) and cell counts method, the description is too simple. What is the sampling frequency? What is the volume of water sample used for qualitative observation and quantitative counting? What are the specific operation steps? Is each quantitative sample counted three times to eliminate the error?

The authors acknowledge the reviewer’s comments, and would like to highlight that specific information are provided in paragraphs 2, 3 and 4 of Section 2.2. Also, the authors have added the relevant reference for the methods used.

  1. Environmental factors in Reservoir C are not presented in this paper, and the nutritional levels of these reservoirs are also unknown.

The authors acknowledge the reviewer’s comments. Limited environmental factors for the catchment are presented at the beginning of Section 3. However, the aim of this work was not to study impact of nutritional levels on bloom formation but to evaluate the impact of sonication on controlling the presence of cells.

  1. There are two “3.2” contents in the result section (3.2. Algal groups timeseries – Reservoir C and 3.2. Cyanobacteria timeseries – Reservoir C), please check and correct.

The authors acknowledge the reviewer’s comments, and the Section 3 subsections have been modified as suggested.

  1. Phytoplankton and cyanobacterial community composition, diversity and richness are also unknown, because I can’t find the specific data. I think that dominant species and rare species should be included in the results.

The authors acknowledge the reviewer’s comments, and would like to highlight that cyanobacterial community is presented in Figure 9 and the Supplementary Information section. However, as the focus of this work was to compare fate of cyanobacteria relevant to other phytoplankton limited community composition were explored and all data is presented in Section 3.

  1. Whether there is a significant difference should be marked at the corresponding position in the text, such as (p<0.05/0.01).

The authors acknowledge the reviewer’s comments, and would like to highlight that definition of significance has been presented in Section 2 and Table 2 in Section 3 is presenting the results.

  1. The presentation of all figures can be improved. And figures 7-9: Latin names of species should be italicized. Some figures can focus on the main contents, such as S1, S5 and S6. I suggest that the data after 2018 should be presented separately. These figures cannot be seen clearly.

The authors acknowledge the reviewer’s comments, and the figures are modified following their suggestion, except for separating 2018 data which the authors believe it needs to be included with all the data as the comparison point for the sonication period.

  1. What's percentage of toxic cyanobacteria in different sites and at different sampling periods? How about the species diversity before and after ultrasonic treatment, was it a single strain or mixed strains? I suggest using a figure or table to make a clear summary.

The authors acknowledge the reviewer’s comments, and would like to highlight that the focus of this study was the main reservoir where the sonicator was installed. The relevant diversity data are presented in Section 3.

Finally, I think this paper didn’t show the specific results of toxins, or whether cyanotoxins were detected in the reservoirs? How about the dynamics of species abundance and toxin-producing before and after Sonicator treatment? I am very interested in this point. I hope the authors can elaborate on the community composition and dynamic changes of harmful cyanobacteria, as well as the concentration and dynamic changes of cyanotoxins.

The authors acknowledge the reviewer’s comments, and while they agree that fate of toxins is an important matter, sonication doesn’t have a direct impact on toxin concentrations. The focus on this study was on impact of sonication on cells as potential toxin producers: Source control measure.  

Reviewer 2 Report

Manuscript: Evaluating Ultrasonicator Performance for Cyanobacteria Management at the source

Comment about the title: Please, be more specific regarding the “source”. The title must be clearer.

General comments: The manuscript highlights the using of an ultrasonicator to control cyanoHABs in situ. Despite the relevance on testing this technology on controlling harmful algal blooms, the case report brings some methodological issues, besides the results which are not presented adequately. Also, some writing issues address some inconsistency which must be carefully revised.

As the manuscript is lacking the lines numbers, it was difficult to indicate exactly where the modifications should be done, so specific comments are made by section.

Abstract

It is more suitable to call "eukaryotic algae" instead of "non-cyanobacterial algae".

Introduction

-          Please, provide a clearer definition to sonication. For instance, how does the sonication remove cells from water column? Also, if cells are lysed following sonication, is there any risk associated to cyanotoxins release in water?

-          Gas vesicles are not organelles, but cell nanocompartiments.

-          Please, consider Raphidiopsis (formerly Cylindropsermopsis).

-          It is suitable to finish the Introduction with a sentence with the main goal(s) of the study.

Methods

-          Figure 1: A map with the watershed as well as the sampling points would be very welcome.

-          Replace “speciation” by "identification".

-          Why only cyanobacterial BV was considered? It cannot be compared with cell density to other phytoplankton groups.

-          Regarding the “quasi-control”, What about the phytoplankton data (at least) of a non-sonicated area during the sonication period?

-          As discussed within this report, these imperfect controls have implications on the reliability of conducted statistical tests.” This sentence limits the evaluation of the technique functioning and weakens the statement that the ultrasound is not efficient to manage CyanoHABs.

-          Table 1: Please, make the information in the table clearer. It was very difficult to understand what you meant. For instance, what about the yellow ones? Maybe it is better to describe that a color gradient from green to red indicate large to smallers datasets, respectively. Also, to which data do you refer to?

Results and discussion

      Please, check if some figures are correctly quoted in the text. For instance, Fig. 3 is mentioned as referring to Fig. 2 and so on. Also, these quotes would be suitable to appear before their figures.

-          Figure 2: Please, use letters (e.g., A; B) to differentiate both graph. Also, I suggest the dataset from all assessed years to be grouped and shown as graph bars (plus standard deviation) for each month, since the seasonal variation is greater than the monthly one.

-          Since Reservoir C is supplied by proximate reservoirs rather than its own catchment, a rainfall event is unlikely to elevate cyanobacteria levels until water from feeder reservoirs, which have greater exposure to agricultural activity and therefore nutrient inputs, reaches Reservoir C. This may have a lengthy and non-constant lag time, which complicates efforts to decouple rainfall variability from algal prevalence.” Ok, but why to show rainfall data? This assumption weakens the relevance of showing meteorological data. As described, the study considers mainly the data on reservoir C, where the technology was applied. Please, be careful.

-          2018 and 2019 were the warmer recorded years (mean max. temperature = 21.8°C) while 2020 and 2021 were cooler (mean max. temperature = 20.4°C and 20.1°C respectively).” Although "warmer" and "cooler" are relative, I think that this should not be considered in the presented temperature mean range (less than 2ºC variation). Furthermore, how could the mean max. temperature be below 22 °C if, e.g., on January temperatures were over 25 °C for all assessed years?

-          It would be very welcome to show if indeed blooms were observed in the warmer period. Characterize the blooms seasons by cyanobacterial biomass, instead of temperature.

-          Please, in the figure’s legends, always remind what means DWTP.

-          Much relevance has been given to the seasonal dynamics of phytoplankton instead of the effects of the ultrasonication.

-          Figure 10: This graph is not much informative to the results section. Maybe it should be moved to the supplementary materials.

-          This sampling did however show a statistically significant reduction in cyanobacteria cell counts during bloom seasons.” This finding reinforces that defining bloom season only by temperature is not correct. Bloom season must be defined by cyanobacterial abundance.

Author Response

Response to Reviewer 2

Manuscript: Evaluating Ultrasonicator Performance for Cyanobacteria Management at the source

Comment about the title: Please, be more specific regarding the “source”. The title must be clearer.

The authors acknowledge the reviewer’s comment, and the tilt has been modified accordingly.  

General comments: The manuscript highlights the using of an ultrasonicator to control cyanoHABs in situ. Despite the relevance on testing this technology on controlling harmful algal blooms, the case report brings some methodological issues, besides the results which are not presented adequately. Also, some writing issues address some inconsistency which must be carefully revised.

The authors acknowledge the reviewer’s comment, and would like to highlight that all used methods were scientific standard procedures and references are provided. All results are also presented within the main text and the supplementary information.

As the manuscript is lacking the lines numbers, it was difficult to indicate exactly where the modifications should be done, so specific comments are made by section.

Abstract

It is more suitable to call "eukaryotic algae" instead of "non-cyanobacterial algae".

The authors acknowledge the reviewer’s comment, and the text has been modified accordingly. 

Introduction

-          Please, provide a clearer definition to sonication. For instance, how does the sonication remove cells from water column? Also, if cells are lysed following sonication, is there any risk associated to cyanotoxins release in water?

The authors acknowledge the reviewer’s comment, and would like to highlight that sonication tech and associated risk have been discussed in details in Section 1.2.

-          Gas vesicles are not organelles, but cell nano compartments.

-          Please, consider Raphidiopsis (formerly Cylindropsermopsis).

-          It is suitable to finish the Introduction with a sentence with the main goal(s) of the study.

The authors acknowledge the reviewer’s comment, and the text has been modified accordingly. 

Methods

-          Figure 1: A map with the watershed as well as the sampling points would be very welcome.

The authors acknowledge the reviewer’s comment, and would like to highlight that sampling points are located in Figure 1. The sonication has no impact at the watershed level hence the map of watershed is not necessary for this paper.

-          Replace “speciation” by "identification".

The authors acknowledge the reviewer’s comment, and the text has been modified accordingly. 

-          Why only cyanobacterial BV was considered? It cannot be compared with cell density to other phytoplankton groups.

The authors acknowledge the reviewer’s comment, and would like to highlight that both cyanobacteria cell number sand BV were explored.

-          Regarding the “quasi-control”, What about the phytoplankton data (at least) of a non-sonicated area during the sonication period?

The authors acknowledge the reviewer’s comment, and would like to highlight that Figure 2 is presented to address this matter.

-          “As discussed within this report, these imperfect controls have implications on the reliability of conducted statistical tests.” This sentence limits the evaluation of the technique functioning and weakens the statement that the ultrasound is not efficient to manage CyanoHABs.

The authors acknowledge the reviewer’s comment, however the authors believe that its imperative to clearly indicate the condition of the trail. Please note that, having access to an identical parallel control water body is practically impossible.

-          Table 1: Please, make the information in the table clearer. It was very difficult to understand what you meant. For instance, what about the yellow ones? Maybe it is better to describe that a color gradient from green to red indicate large to smallers datasets, respectively. Also, to which data do you refer to?

The authors acknowledge the reviewer’s comment, and would like to highlight that the color codes were generated to clarify the take home message.

Results and discussion

      Please, check if some figures are correctly quoted in the text. For instance, Fig. 3 is mentioned as referring to Fig. 2 and so on. Also, these quotes would be suitable to appear before their figures.

The authors acknowledge the reviewer’s comment, and the text has been modified accordingly. 

-          Figure 2: Please, use letters (e.g., A; B) to differentiate both graph. Also, I suggest the dataset from all assessed years to be grouped and shown as graph bars (plus standard deviation) for each month, since the seasonal variation is greater than the monthly one.

The authors acknowledge the reviewer’s comment, however they would like to highlight that these graphs are presented only for broad discussion about the environmental situation.  

-          “Since Reservoir C is supplied by proximate reservoirs rather than its own catchment, a rainfall event is unlikely to elevate cyanobacteria levels until water from feeder reservoirs, which have greater exposure to agricultural activity and therefore nutrient inputs, reaches Reservoir C. This may have a lengthy and non-constant lag time, which complicates efforts to decouple rainfall variability from algal prevalence.” Ok, but why to show rainfall data? This assumption weakens the relevance of showing meteorological data. As described, the study considers mainly the data on reservoir C, where the technology was applied. Please, be careful.

The authors acknowledge the reviewer’s comment and agree, hence the rainfall data were presented only for broad discussion about the environmental situation.  

-          “2018 and 2019 were the warmer recorded years (mean max. temperature = 21.8°C) while 2020 and 2021 were cooler (mean max. temperature = 20.4°C and 20.1°C respectively).” Although "warmer" and "cooler" are relative, I think that this should not be considered in the presented temperature mean range (less than 2ºC variation). Furthermore, how could the mean max. temperature be below 22 °C if, e.g., on January temperatures were over 25 °C for all assessed years?

The authors acknowledge the reviewer’s comment, and the text has been modified accordingly. 

-          It would be very welcome to show if indeed blooms were observed in the warmer period. Characterize the blooms seasons by cyanobacterial biomass, instead of temperature.

The authors acknowledge the reviewer’s comment, and would like to highlight that in these water bodies blooms can occur at different water temperatures. . 

-          Please, in the figure’s legends, always remind what means DWTP.

The authors acknowledge the reviewer’s comment, and the text has been modified accordingly. 

-          Much relevance has been given to the seasonal dynamics of phytoplankton instead of the effects of the ultrasonication.

The authors acknowledge the reviewer’s comment, and would like to highlight that discussing seasonal dynamics of phytoplankton is important as previous studies have associated low cell numbers to sonication impact while in general the seasonal dynamics were the reason behind lower cell numbers.

-          Figure 10: This graph is not much informative to the results section. Maybe it should be moved to the supplementary materials.

The authors acknowledge the reviewer’s comment, however they believe its important to emphases the importance of statistical analysis.

-          “This sampling did however show a statistically significant reduction in cyanobacteria cell counts during bloom seasons.” This finding reinforces that defining bloom season only by temperature is not correct. Bloom season must be defined by cyanobacterial abundance.

The authors acknowledge the reviewer’s comment, the bloom season was defined using both cell numbers and temperature, but in some cases due to low cell numbers only temperature data were used to define the threshold.

Reviewer 3 Report

Manuscript refers on alternative methods such as sonication on controlling cyanobacterial cell densities. Though the manuscript carries relevant information on carries a lack on clarifications. To start how many sonicators are on the market and why was the one tested chosen? Also was this sonicator applied to other studies or is this a new description? Authors should clarify this.

Another remark refers on how were the cell counts achieved? Were by traditional microscopy methods? There should be a description in the Methods section.Does the trophic level on ecosystems contribute to the same response on application on sonicator? I mean were the water tested on this sonicator on the literature and on this trial eutrophic?

Finally, what are the new insights on this trial on comparison on literature on application on this device? I mean if there was validation to be on the market what are the parameters on this reservoir that influence this down-response on the sonicator?  

Author Response

Response to Reviewer 3

Manuscript refers on alternative methods such as sonication on controlling cyanobacterial cell densities. Though the manuscript carries relevant information on carries a lack on clarifications. To start how many sonicators are on the market and why was the one tested chosen? Also was this sonicator applied to other studies or is this a new description? Authors should clarify this.

The authors acknowledge the reviewer’s comment, however, they would like to highlight that the aim of this work was not to compare commercial products but to evaluate the technology. This Sonicatr is currently being used in other ongoing studies and past published studies are cited within this paper.

Another remark refers on how were the cell counts achieved? Were by traditional microscopy methods? There should be a description in the Methods section. Does the trophic level on ecosystems contribute to the same response on application on sonicator? I mean were the water tested on this sonicator on the literature and on this trial eutrophic?

Cell counts were achieved using standard microscopy taxonomy and the refence is cited within the paper. The tested water between this study and other studies were compared; however, the available information is very limited.

Finally, what are the new insights on this trial on comparison on literature on application on this device? I mean if there was validation to be on the market what are the parameters on this reservoir that influence this down-response on the sonicator?

The authors acknowledge the reviewer’s comment, and they would like to highlight that in this study number of sonicators were matched to the size of water body and proper cell count were conducted. As mentioned in the text the technology limitations and also operational malfunction contributed to the results as well.  

Author Response

Response to Reviewer 4

The paper was about testing the presence of a sonicator in drinking water reservoir at Victoria, Australia. However, based on the presented data no strong evidence of a change in cyanobacteria densities was observed but minimal disruptions to the dominant cyanobacteria species was detected.

To improve the quality of the paper here are some comments:

Introduction:

  • Was the reservoir susceptible to algal blooms and any record of bloom was reported? If the response if positive what’s the dominant species that created bloom in this reservoir?

The authors acknowledge the reviewer’s comment; yes the reservoir was susceptible to algal blooms and bloom were observed in the past, only visual observation. The speciation results are presented in Section 3.

Material and Methods:

  • What was the volume of reservoir? How deep the reservoir was? What was the geometry of the reservoir?

The reservoir maximum depth was 6m. As indicated in the paper the sonicators were selected and installed to address the geometry conditions.

  • Was there any barrier or filter between each reservoir to the reservoir C?

No there was no filtration barrier between reservoirs.

  • In page 4 section 2.2. the authors claimed that “‘DWTP intake’ sampling was conducted from this pipe to provide an accurate representation of water quality abstracted from the reservoir and entering the treatment plant”. How accurate was it? Did the authors samples from any other sections of reservoir other than the ones mentioned, including location closed to the sonicator?

The authors acknowledge the reviewer’s comment; all 3 sampling locations are presented in Figure 1. Sampling the at DWTP intake is indeed the most representative sample of the water post sonication that leave the reservoir for treatment and consumption.

  • Page 5 the first line under Figure 2 doesn’t need indent.

The authors acknowledge the reviewer’s comment, and the text is now corrected accordingly.

  • Table 1. I recommend using a color band. A color band would be a better option to show the difference between low and high sampling.

The authors acknowledge the reviewer’s comment, however the limited color codes were used to keep the message simple and easy to follow.

  • What was the power and pulse duration of the sonicator in this experiment? Was it used continuously or not?

The sonicator was used continuously apart from the period of malfunction which was one of the sources of performance issues.

  • Sonication ruptures the cells which can release the toxin(s) into water. Did the authors consider this? Did you do any toxicology test to measure the level of toxins in the water and be sure about the quality of the drinking water?

The authors acknowledge the reviewer’s comment; yes the author conducted toxin analysis and no toxins were detected in the water. However, the focus on this study was on fate of cell during sonication.

Results and Discussion

  • What was reservoir C temperature? Did the author measure pH or other water parameters during the experiment?

The authors acknowledge the reviewer’s comment. As mentioned these water quality parameters were measured by the YSI probe however due to calibration issues where not included.

  • When the sonicator introduced for the first time to the reservoir was there any bloom available? If the response was positive which stage of the bloom, it was and what was the dominant species?

The authors acknowledge the reviewer’s comment. No at the time of installation no visible bloom was detected in the reservoir.

  • Page 12; it is mentioned: “An ex-ample is provided in Figure 12.” There is no Figure 12. The total Figures are 10.

The authors acknowledge the reviewer’s comment, and the text is now corrected accordingly.

  • Were there any sample replicates for cell count and biovolume?

Yes, all samples were taken in triplicate following standard methods.

  • The discussion can be improved by adding and comparing with similar works.

The authors acknowledge the reviewer’s comment and would like to highlight that comparison has been made with published results; however, information on trial conditions are limited. 

Round 2

Reviewer 1 Report

I consider that the manuscript was improved with the adjustments made. The authors followed the suggestions and comments of the reviewers. However, there are still some questions in the manuscript that have not been answered correctly. I would recommend some points:

1. “Where is the specific location of the site “reservoir water”? Please provide.

Is the water inside the reservoir and each reservoir has been identified separately (Figures 1 and 2).”

So do you mean that the reservoir water samples in each reservoir are not collected at fixed location every time?

2. “What is the specific detection method of DWTP intake, including physical and chemical parameters, phytoplankton and cyanobacterial counts, etc.

The authors acknowledge the reviewer’s comments and would like to highlight that same detection methods was used for all sampling sites as indicated within the manuscript.”

Of course, all sampling sites should use the same detection methods, but what I want to know is the specific methods. Please provide or add appropriate references.

3. For cyanobacteria biovolumes (BV) and cell counts method, I dont think that the specific information are all provided in paragraphs 2, 3 and 4 of Section 2.2. So what is the sampling frequency? What is the volume of water sample used for qualitative observation and quantitative counting? What are the specific operation steps? Is each quantitative sample counted three times to eliminate the error? etc.

I would like to suggest that the authors use some words to explain clearly us rather than simply add a reference on the basis of the original text.

4. Figures 7-9: Latin names of species should be italicized. I suggest only Latin names of genera and species are italicized, sp. and “Aphanizomenonaceae Family-Straight” are not italicized.

Finally, I want to thank the authors for their good work.

Author Response

1. “Where is the specific location of the site “reservoir water”? Please provide.

Is the water inside the reservoir and each reservoir has been identified separately (Figures 1 and 2).”

So do you mean that the “reservoir water” samples in each reservoir are not collected at fixed location every time?

Authors: Water samples were collected separately and at fixed location and time. 

2. “What is the specific detection method of DWTP intake, including physical and chemical parameters, phytoplankton and cyanobacterial counts, etc.

The authors acknowledge the reviewer’s comments and would like to highlight that same detection methods was used for all sampling sites as indicated within the manuscript.”

Of course, all sampling sites should use the same detection methods, but what I want to know is the specific methods. Please provide or add appropriate references.

Authors: References are added.

3. For cyanobacteria biovolumes (BV) and cell counts method, I don’t think that the specific information are all provided in paragraphs 2, 3 and 4 of Section 2.2. So what is the sampling frequency? What is the volume of water sample used for qualitative observation and quantitative counting? What are the specific operation steps? Is each quantitative sample counted three times to eliminate the error? etc.

I would like to suggest that the authors use some words to explain clearly us rather than simply add a reference on the basis of the original text.

Authors: Reference and explanation added. 

4. Figures 7-9: Latin names of species should be italicized. I suggest only Latin names of genera and species are italicized, “sp.” and “Aphanizomenonaceae Family-Straight” are not italicized.

Author: Due to font issues decided to make all caption in these figures italicized. 

Reviewer 2 Report

Dear Authors, 

Although the manuscript's subject is interesting for Toxins readers, most of the pointed critical issues were not adequately modified/clafiried and some flaws still remain, besides the lack of line numbers which have difficulted to precisely indicate the necessary changes.

For instance:
- 6 months is a very long temporal control. However, it would be suitable to assume a quasi-control on a point previous to the area of sonication influence. How can you argue that the previous phytoplankton structure was different due to the sonication absence and not other water changes? It is important to assume control area and at least temporal replicates with a shorter time-scale (a few days or a week) considering that the phytoplankton might change rapidly.

It still not clear what you mean with the table 1. Please, clarify or remove it. What take home message do you mean? It seems a table with the number of samples per area.

Quotes remains after the figures;

Please, consider a broad discussion about the environmental situation to help to understand the data and not to weaken them;

A bloom season is mentioned, however seems like there was no bloom record since even under relatively warmer periods, a low cyanobacterial cell density was recorded.

Author Response

6 months is a very long temporal control. However, it would be suitable to assume a quasi-control on a point previous to the area of sonication influence. How can you argue that the previous phytoplankton structure was different due to the sonication absence and not other water changes? It is important to assume control area and at least temporal replicates with a shorter time-scale (a few days or a week) considering that the phytoplankton might change rapidly.

Authors: We clearly indicated in the manuscript that we did not have access to physical control. It was not possible to built a full size reservoir with exactly same conditions. I don't believe it would be possible at all to have such ideal control. Hence we have used 3 years of historic data and we have conduced the trial for 2 bloom seasons. We have also discussed our results being mindful of these limitations. 

It still not clear what you mean with the table 1. Please, clarify or remove it. What take home message do you mean? It seems a table with the number of samples per area.

Authors: Exactly Table 1 is to show the extend of sampling work.   

Please, consider a broad discussion about the environmental situation to help to understand the data and not to weaken them;

Authors: Section 3.1 is allocated to this discussion. 

A bloom season is mentioned, however seems like there was no bloom record since even under relatively warmer periods, a low cyanobacterial cell density was recorded.

Authors: We agree with the reviewer; with bloom season we were referring to the warmer period of year as this is how the utilities classify the year and apply their management strategy. 

Reviewer 3 Report

Authors on adressed on comments and now manuscript warrants publication on Toxins.

Author Response

Authors on addressed on comments and now manuscript warrants publication on Toxins.

Authors: Many thanks for the constructive comments and support.